# Age-Related Skeletal Muscle Dysfunction Is Aggravated by Obesity: An Investigation of Contractile Function, Implications and Treatment

**DOI:** 10.3390/biom11030372

**Published:** 2021-03-02

**Authors:** Jason Tallis, Sharn Shelley, Hans Degens, Cameron Hill

**Affiliations:** 1Centre for Applied Biological and Exercise Sciences, Alison Gingell Building, Coventry University, Priory Street, Coventry CV15FB, UK; shelley3@uni.coventry.ac.uk; 2Research Centre for Musculoskeletal Science & Sports Medicine, Department of Life Sciences, Manchester Metropolitan University, Manchester M15 6BH, UK; h.degens@mmu.ac.uk; 3Institute of Sport Science and Innovations, Lithuanian Sports University, 44221 Kaunas, Lithuania; 4Randall Centre for Cell and Molecular Biophysics, New Hunt’s House, Guy’s Campus, King’s College London, London SE1 1UL, UK; cameron.hill@kcl.ac.uk

**Keywords:** high-fat diet, sarcopenia, muscle ageing, isolated skeletal muscle

## Abstract

Obesity is a global epidemic and coupled with the unprecedented growth of the world’s older adult population, a growing number of individuals are both old and obese. Whilst both ageing and obesity are associated with an increased prevalence of chronic health conditions and a substantial economic burden, evidence suggests that the coincident effects exacerbate negative health outcomes. A significant contributor to such detrimental effects may be the reduction in the contractile performance of skeletal muscle, given that poor muscle function is related to chronic disease, poor quality of life and all-cause mortality. Whilst the effects of ageing and obesity independently on skeletal muscle function have been investigated, the combined effects are yet to be thoroughly explored. Given the importance of skeletal muscle to whole-body health and physical function, the present study sought to provide a review of the literature to: (1) summarise the effect of obesity on the age-induced reduction in skeletal muscle contractile function; (2) understand whether obesity effects on skeletal muscle are similar in young and old muscle; (3) consider the consequences of these changes to whole-body functional performance; (4) outline important future work along with the potential for targeted intervention strategies to mitigate potential detrimental effects.

## 1. Introduction

Obesity is a global epidemic [1] associated with poor physical and mental health, reduced quality of life, and increased morbidity and mortality [2,3,4,5]. More specifically, obesity has been linked to increased risk of cardiovascular disease, insulin resistance, non-alcoholic fatty liver disease, subfertility and cancer [5]. Obesity is also recognised to have substantial economic repercussions as a result of reduced productivity, unemployment and direct health care costs [6]. In 2014, the global economic impact of obesity was estimated to be USD 2 trillion, or 2.8% of the global gross domestic product (GDP) [7].

The world is faced with unprecedented growth in the older adult population [8], which is accompanied by an increased prevalence of chronic diseases and hence a substantial economic cost for health care [9,10]. Data from the US indicate a growing trend in the prevalence of obesity in adults aged >60 years, with over 37% of this population being obese [11]. Recent data from the UK demonstrate that the proportion of adults that are obese increases with age, with ~30% of adults >65 years being classified as obese [12]. Given the growth in the world’s old obese population, there has been a recent surge in literature examining the combined effects of increasing age and obesity on markers of health and physiological function. When compared to the independent effects of increasing age and obesity, old obese adults have an increased risk of cardiovascular disease (CVD), metabolic disorders, and all-cause mortality [13,14,15,16,17]. One important target for investigation has been skeletal muscle. Skeletal muscle is the largest regulator of metabolism in the body, and effective contractile function is required for locomotor performance and physical activity [18]. More specifically, muscle weakness is independently related to fall risk, chronic disease, poor quality of life and greater all-cause mortality [19,20,21,22]. Whilst the effects of age on contractile function are well established [23,24,25,26], there is growing evidence that obesity also has a negative effect on muscle function that may in turn act as a catalyst to a negative obesity cycle [27], thus exacerbating the detrimental effects of high adiposity on physiological function and health in old age. Despite these suggestions, evidence for the negative effects of obesity and ageing on the contractile function of skeletal muscle appear limited and controversial.

## 2. Defining Skeletal Muscle Contractile Function

Some of the ambiguity in understanding the direct effects of obesity on skeletal muscle contractile function are linked specifically to the definition of muscle function. As such, in the context of this review, it is important to define this at the outset. Whilst previous work in this area has employed a variety of methodological procedures, skeletal muscle contractility has primarily been assessed in three ways. Most studies focus on the absolute maximal strength (force output) or power (force x velocity of movement) -producing capacity of a muscle or muscle group [28,29,30] and other studies have also examined the contractile function normalised to body mass (i.e., force to body mass) [31,32,33,34]. Such measures are important in understanding the biomechanical consequences of changes in muscle function and limits to functional performance. To a much lesser extent, previous work has examined the maximal force or power output normalised to muscle size (typically cross-sectional area or muscle mass) as an indicator of muscle quality [34,35,36]. These latter studies suggested an obesity-induced reduction in the intrinsic force-producing capacity of skeletal muscle, or muscle quality, and explains the lower force or power despite larger muscles in the obese [18,37]. In addition, larger muscles of poorer quality also result in greater body inertia and an increased cost of maintenance, repair and regeneration [27]. Each of these measures not only provides valuable insight into understanding of the direct impact on skeletal muscle function but also allows inferences with respect to functional consequences. As such, in this review, we will consider the effects of age, obesity and their potential additive effect on each of these aspects of skeletal muscle function.

The combined effects of ageing and obesity on skeletal muscle are commonly referred to as sarcopenic obesity [13,14,38]. Sarcopenia, defined as an age-related reduction in muscle mass and an associated reduction in contractile function [25], is commonly referred to in the published literature exploring the effects of age on skeletal muscle function [24,25,26,39,40,41,42]. In the context of the current review, we choose not to refer to the term sarcopenic obesity. Skeletal muscle ageing can occur without substantial atrophy [43,44], and in many of the studies examining the effects of both ageing and obesity on contractile function, skeletal muscle mass is not measured [31,32] or is unchanged or increased [45,46].

## 3. The Independent Effects of Increasing Age and Obesity on Skeletal Muscle Contractile Function

Despite the complexities associated with understanding the muscle-specific ageing response, an age-induced reduction in the contractile performance of skeletal muscle has been firmly established [25,42,47], and strongly linked to reduced balance, reduced mobility and all-cause mortality [19,48,49,50]. An age-related reduction in muscle mass has been shown to occur as early as 25 years of age and is substantially lower in later life [51]. Muscle ageing is typically characterised as sarcopenia [25]; however, the loss of muscle mass and function is disproportionate, with the reduction in contractile function exceeding the loss of muscle mass or muscle cross-sectional area [52,53,54,55], with this phenomenon termed dynapenia [39]. In fact, the loss of strength has been shown to occur 2–5 times faster than the loss of mass [42], indicating that age-related muscle atrophy is prevalent much later in the ageing process and likely further accelerates the loss of muscle function.

More specifically increasing age is associated with a reduction in skeletal muscle quality [43,44,56], maximal force-generating capacity [24] and shortening velocity [47], which typically combine to result in a more severe loss of muscular power [57,58,59]. Muscular power is of greater importance to the completion of activities of daily living than muscle strength [60], with muscular power more strongly correlated with age-adjusted all-cause mortality than muscular strength [61]. Such changes in the above-mentioned contractile properties are evident across postural, locomotor and respiratory muscles [24,62,63], although the magnitude of the response is influenced substantially by numerous factors including physical inactivity, diet and disease [64,65]. One such confounding factor is obesity, with evidence to suggest that excessive adipose tissue accumulation may exacerbate the age-related decline in muscle function [34,45,46].

Although to a relatively much smaller extent, and with greater ambiguity in findings, there is a growing body of evidence to suggest that obesity independent of age results in a muscle-, sex- and contractile mode-specific effect on skeletal muscle function [27,28,29]. Whilst a similar argument can be made for evidence examining the muscle ageing response [43,44,56,66,67,68,69], experimental models examining the effects of high-fat diet (HFD) consumption on isolated skeletal muscle function have been important in developing our understanding of the effects of obesity on muscle structure [70] and function [18,37,71,72]. This method bypasses the impact of intramuscular fat during obesity and the effects of age on the neuromuscular activation of skeletal muscle [73] and thus allows one to assess muscle-specific effects with more precise control of HFD nutritional composition and feeding duration [27], important factors that differ significantly between obese people. Furthermore, studies employing isolated skeletal muscle models allow an accurate assessment of muscle quality, where contractile performance can be normalised to cross-sectional area (CSA) or mass of the muscle examined. Assessments of contractile function normalised to whole-body or segmental lean mass, typically as per the approaches adopted in whole-body in vivo testing in humans [31,34,36,74], have likely resulted in conflicting conclusions with respect to obesity effects on muscle quality [27].

When the literature is considered holistically, and although highly specific, previous work has indicated evidence for an obesity-induced increase in the absolute force-producing capacity of antigravity muscles, a reduction in force to body mass ratio and a reduction in muscle quality [18,75,76]. Muscles with a faster fibre type composition likely show greater detrimental effects [18,37], with the response further dependant on the nutritional composition of the diet [71] and magnitude of adiposity. For example, Hurst et al. [37] examined the effect of 2, 4, 8 and 12 weeks of HFD feeding on the contractile performance of isolated mouse soleus, EDL or diaphragm muscle. Whilst muscle and contractile mode-specific detrimental effects of HFD on measures of muscle quality and fatigue resistance were seen only after 8 weeks, correlations between gonadal fat pad mass and measures of contractile function indicated that irrespective of feeding duration, level of adiposity was an important factor dictating the magnitude of the HFD-induced response.

## 4. Age- and Obesity-Induced Mechanistic Changes Related to Altered Skeletal Muscle Contractile Function

Based on the independent effects of ageing and obesity outlined above, one would expect that ageing and obesity together would exacerbate decrements in skeletal muscle function. This hypothesis is strengthened by examining the mechanistic characteristics associated with age- and obesity-induced changes in skeletal muscle function. Both have been robustly examined in previous reviews of the literature [23,25,26,27,77,78,79,80,81] and will be summarised here. Given the muscle- and contractile mode-specific nature of both the muscle ageing response and the effects of obesity, mechanisms underpinning these changes are likely also muscle specific.

The age-related reduction in skeletal muscle function is multifactorial [23]. Some of the contributing factors are a reduction in the sarcoplasmic reticulum (SR) Ca^2+^ concentration, dihydropyridine receptor–ryanodine receptor uncoupling (i.e., excitation–contraction uncoupling) and reduced sarco(endo)plasmic reticulum Ca^2+^-ATPase (SERCA) activity [43,82,83] that all have a negative impact on muscle activation and relaxation. A reduced myogenic capacity and an increase in catabolic agents [23,84] result in a reduction in contractile mass [52,53,54], particularly in fast fatigable muscle fibres [85]. Furthermore, ageing is associated with a reduced mitochondrial function [86] and increased heterogeneity of capillary spacing [87].

A HFD induced changes in muscle function similar to that seen in muscle ageing, with substantial HFD consumption linked to chronic low-grade inflammation and a reduction in myogenesis [77,88,89,90,91], impaired excitation–contraction coupling [71,92], and a reduction in mitochondrial ATP regeneration [93]. However, HFD consumption has been linked to a shift towards a faster muscle phenotype [27], which appears to be at odds with the typical ageing response.

Based on the independent effects of ageing and obesity, there appears to be a strong rationale to support the hypotheses that obesity has the potential to aggravate the age-related reduction in skeletal muscle function. Herein, we provide a review of the evidence that has sought to address this question.

## 5. Effects of Ageing and Obesity on Skeletal Muscle Contractile Function

Evidence suggests that individuals that are both old and obese have compromised function in activities of daily living when compared to normal weight counterparts [2,94,95]. Whilst this in part may be attributed to kinematic differences brought about through local adipose tissue storage [96], joint pain [97], and impaired neurocognitive function [98], a significant contributor to these changes is likely related to the increased demand placed on skeletal muscle. Evidence examining the potential additive effects of ageing and obesity on the muscle function of older adults is limited and controversial (see Table 1).

Similar to observations in young obese individuals [75], there is evidence that obesity may also result in an increased absolute force of locomotor and postural muscles of older adults [99,104], whilst non-antigravity muscles are unaffected [99]. This increase in absolute muscle force-generating capacity has been attributed to increased loading of the musculoskeletal system as a result of supporting and ambulating a greater load [27]. Other studies, however, show that in older adults there is no such obesity-related increase in absolute force-generating capacity [31,33,100,105,106] and in some cases, obesity has even been reported to result in a reduced absolute force-producing capacity of the musculature of older adults [36,105]. The disparity in response between young and old obese groups may in part be explained by an age-induced reduction in myogenesis [107], limiting the adaptations that may occur through elevated loading.

Whatever happens with the absolute force-generating capacity, like in young adults, there is evidence that the force to body mass ratio is significantly lower in obese older adults compared to normal weight counterparts [31,33]. A reduction in the force to body mass ratio will have effects that are more substantial for older adults where functional mobility may be already compromised. Conversely, Tomlinson et al. [100] rejects these trends indicating no difference in peak isometric force of the plantar flexors muscles relative to body mass between old obese and age-matched lean equivalents, concluding that there may be a potential protective effect in old compared to young obese.

Given that both ageing and obesity effects on skeletal muscle function are likely influenced by several factors, a lack of homogeneity in the sampled participants may account for the disparity in results. For example, Tomlinson et al. [100] demonstrated that obese aged participants displayed a trend for an increase in peak torque of the plantar flexors and reduced force to body mass ratio at a level that did not reach statistical significance. Both ageing and obesity responses are likely influenced by physical activity, diet, comorbidities, regional adipose tissue accumulation and magnitude and duration of adipose tissue accumulation [27,64,65,108,109], which may account for a high degree of variation in the population sampled and makes it difficult to make direct comparisons between published work. Furthermore, summarising current evidence is challenging given the broad range of ages examined (58–80 yrs), the limited focus on older and geriatric age groups and a disproportionate focus on female participants compared to males.

It is firmly established that reduced physical activity contributes to the age-related decline in contractile performance [24]. Given that obesity may result in a decrease in physical activity levels [110], it is a challenge to distinguish between obesity-induced changes in contractile function and those that result from physical inactivity. Surprisingly, few studies examining the additive effects of age and obesity have measured or controlled for physical activity. When adjusted for height, physical activity, pain, depression, and muscle mass, Rolland et al. [99] demonstrated that in older women, obesity-induced increases in hand grip strength, peak isometric force of the elbow extensors and peak isometric force of the knee extensors for the sedentary group, were no longer apparent. However, higher peak isometric torque of the knee extensors was still prevalent in the obese active group compared to the active lean control. Interestingly, Paolillo et al. [33] also indicated no difference in peak concentric torque of the quadriceps muscles in obese postmenopausal women compared to a normal weight group matched for age and anaerobic capacity.

Based on the presented evidence, an aggravated decline in the functional performance of old obese individuals may be explained by a reduction in the force to body mass ratio, what is less clear from the available evidence is the impact of obesity on the intrinsic force-producing capacity of muscle. In vivo studies examining obesity effects of muscle quality have resulted in ambiguous findings [28,30] and studies specifically examining the additive effects of obesity and ageing are sparse. There is some evidence to indicate that muscle performance normalised to whole-body or regional lean mass is significantly reduced in old obese adults [33,106]. Using a biopsy of the vastus lateralis, Choi et al. [34] demonstrated that power normalised to fibre CSA was significantly reduced in type I fibres of obese older adults when compared to normal-weight controls. Such evidence supports a muscle fibre type-specific, obesity-induced reduction in muscle quality.

## 6. Effects of Ageing and Obesity on Contractile Function of Isolated Rodent Skeletal Muscle

Whilst there is a growing body of work examining the effects of HFD on isolated skeletal muscle function [27], only a small number of studies have examined the combined effects of ageing and HFD (see Table 2 for a complete summary of results). Abrigo et al. [111] demonstrated that 12-week old male C57BL/10J mice fed a HFD for 38 weeks had lower absolute strength and strength relative to body mass when assessed in vivo. Furthermore, the HFD fed mice had reduced specific force (force relative to muscle CSA) of the tibialis anterior across a range of stimulation frequencies. By contrast, Bott et al. [112] showed that the consumption of a HFD for 13 weeks did not significantly diminish the specific force of EDL and soleus isolated from 33-week-old male C57BL/6J mice when compared to age-matched controls. HFD consumption did, however, result in increased CSA of soleus, supporting the idea of a HFD-induced hypertrophy of the postural muscles. Whilst the disparity in findings for measures of specific force may be accounted for by substantial differences in HFD feeding duration, it should also be considered that the aged model used by Abrigo et al. [111] and Bott et al. [112] might not be the most suitable for determining the additive effects of HFD consumption and ageing on skeletal muscle function. The 50- and 33-week-old mice used in these studies may represent an early ageing response [43] and are much younger than 18–24 months age groups typically used in studies examining effects of muscle ageing [113]. In the work by Bott et al. [112], other than an age-induced increase in isometric relaxation time, ageing had no other effects on contractile function of the EDL. As such, these findings may not reflect the effects of HFD consumption on the contractile performance of muscles extracted from older animals.

Hill et al. [45] later demonstrated that HFD consumption aggravated the age-related decline in the contractile function of respiratory muscle with limited effects on locomotor muscle performance (Table 2). Using 79-week-old female CD-1 mice, a model that has a substantial age-related decline in muscle function [56], 9 weeks HFD resulted in an increased muscle mass and maximal absolute work loop power for whole soleus and EDL muscle. Peak isometric force, specific force, work loop power normalised to muscle mass and fatigue resistance were unchanged. These findings would appear to, in part, contradict previous work using a similar methodological approach to examine skeletal muscle contractile performance in young HFD-fed rodents [18,37]. In contrast to the EDL muscle, isometric stress and work loop power normalised to muscle mass were reduced in the diaphragm of HFD group, indicating that obesity caused a muscle-specific aggravation of the typical ageing response. A reduction in diaphragm function may have consequences for lung function and respiratory disease risk [114,115]. Whilst this work offers insight into the combined effects of age and obesity, generalisability of these data should be treated with caution. The large changes in body composition over a relatively short time period may not represent the pattern of changes in body composition over the life course. Furthermore, whilst the body mass and gonadal fat mass (FM) were larger in HFD than control mice, the control group used in this study had substantially greater adiposity than controls used in comparable young rodent studies as a result of the typical ageing process [18,37]. While it is not possible to rule out that obesity did not contribute already to the age-related decline in muscle function seen in the control group, it does show that additional HFD-induced increases in FM may be detrimental to the contractile performance of specific skeletal muscle. Based on previous work in young rodent models, it may be assumed that the demonstrated obesity effects would be more substantial if comparisons were made to a lean control [37].

Eshima et al. [46], examining the effect of 20-month HFD feeding on the contractile performance of whole soleus and EDL muscle from male C57BL6J mice, addresses some of these issues. Chronic HFD consumption and the associated increase in body mass more likely to reflect those seen in obese humans. Furthermore, the lean control group used as a comparator did not display a significantly different body mass or abdominal visceral FM to that of a younger 6-month-old lean control group. Whilst an increase in age resulted in a decrease in specific force for the EDL across a range of stimulation frequencies, both specific and absolute force of the 20-month HFD-fed mice was reduced compared to the age-matched lean control. Such trends were similar to EDL isolated from 4-week HFD-fed mice when compared to an age-matched lean equivalent, with the HFD-induced reduction in contractile function in part attributed to dysfunction in intracellular Ca^2+^ release. Conversely, neither age nor HFD significantly influenced the contractile performance of the soleus muscle, similar to previous work [45]. Such findings for the soleus muscle are difficult to evaluate in the context of potential additive effects of both ageing and increased adiposity, as there was no age-associated reduction in the soleus performance.

Although the varied methodological approaches and likely sex, age, HFD-feeding duration and contractile mode-specific response [18,27,37,46,116] make direct comparisons challenging, evidence indicates that HFD consumption exacerbates the age-related decline in muscle function and that the HFD response in older muscles may differ to that seen in younger counterparts. Limited evidence makes it difficult to determine if the severity of excessive adipose tissue accumulation on muscle function is either greater or less substantial in aged muscles. The only study to make direct comparisons between HFD effects on isolated muscle function from young and old mice indicates that these were similar in young and old rodents [46]. When considered collectively, work from the Tallis et al group indicates that HFD consumption is more detrimental to locomotor muscle quality at younger age groups compared to older, whilst HFD effects on the diaphragm are similar [18,37,45].

## 7. Sex-Specific Effects of Obesity on the Age-Related Loss of Skeletal Muscle Contractile Function

The age-associated reduction in muscle contractile function is apparent in both men and women. It is clear, however, that at peak physiological maturity, men typically show higher levels of absolute power, power relative to body mass and specific power than women [117]. There is some suggestion that the absolute and relative loss of absolute and specific contractile function is faster in men compared to women [117]. While the general consensus is indeed that the age-induced loss of absolute strength [52,54,118,119,120] and power [57,121] is greater for men than women, it should be noted that the relative decline is similar and that the larger absolute decline in men is just a consequence of the higher starting point [59] In line with this, studies that show no difference in the magnitude of the decline in absolute force and power between males and females [55,118,122,123] are talking about the relative decline in muscle function. For example, McPhee et al. [55] showed that both younger and older men had greater absolute knee extensor torque and physiological cross-sectional area (PCSA) of the quadriceps than women, the relative magnitude of the decline in contractile function and quadriceps PCSA was similar between men and women.

Likewise, there is little evidence for sex-specific effects of obesity-induced changes in contractile function of aged muscle. In young adults, obesity induces sex-specific changes in FM: fat-free mass (FFM) ratio. Lafortuna et al. [76] reported that there is a concomitant increase in the FM and FFM in obese men, while in obese women FM increased but changes FFM were limited. Despite these differences, obesity-induced changes in muscle function were comparable in men and women. Lafortuna et al. [124] further demonstrated that after adjusting for age, muscle volume increased to a greater extent in obese males compared to female counterparts. However, only Miyatake et al. [31] has directly compared the sex-specific effect of obesity on aged skeletal muscle function, demonstrating that obesity effects of grip strength and absolute and normalised knee extensor strength were comparable between males and females.

Irrespective of age, previous work examining the sex-specific effects of HFD on isolated skeletal muscle contractile function are also sparse. For the most part, studies are constrained to a single-sex and comparisons between published works is challenging given the varied methodological approaches [27]. No study has yet considered the sex-specific effects of HFD and ageing on isolated skeletal muscle function (Table 2). It is therefore not yet possible to robustly determine if the effects of obesity on the muscle ageing response are sex specific, highlighting an important area for future investigation.

## 8. Consequences for Functional Performance

Unlike in human experiments, there is a much clearer trend for an increase in absolute force and power-producing capacity of postural and locomotor musculature in studies that have examined the additive effects of ageing and HFD in rodent isolated muscle [45,46]. However, this seemingly positive adaptation does not correspond with the magnitude of the increase in body mass [45,46]. When the available evidence is considered holistically, both a reduction in force to body mass ratio and muscle quality likely contribute to the impaired functional performance seen in obese older adults [125,126,127]. More specifically, old obese individuals have been shown to have reduced postural control, a slower more tentative gait pattern, increased knee joint loading and altered movement patterns during the completion of activities of daily living, potentially exacerbating musculoskeletal injury risk [128,129,130,131,132,133]. Whist these biomechanical differences may be related to kinematic changes brought about via a change in the position of the centre of mass and restricted joint range of motion [134], the evidence presented here indicates that an accelerated obesity-induced reduction in skeletal muscle function is likely a substantial contributor to these changes. We have summarised the effects of old age and obesity on the contractile performance of locomotor muscles in Figure 1. We have also considered how this contributes to the negative cycle of obesity, whereby poorer contractile function reduces the ability to perform activities of daily living, which in turn reduces the level of physical activity in old obese adults and leads to further reductions in contractile performance, with the overall effect being a reduced quality of life.

Compared to normal-weight counterparts, old obese individuals may be at greater risk of musculoskeletal injury given the importance of adequate muscle function for maintaining joint stability and absorbing impact forces [135,136,137]. For example, old obese individuals have greater fall risk [138,139], with muscular strength recognised as an important facet of postural control [140,141]. Furthermore, the impact of musculoskeletal injury may be more severe in old obese adults, as discussed in more detail below.

Those with age-induced impaired muscle function and obesity have been shown to have a greater risk of metabolic syndrome, cardiovascular disease, diabetes and all-cause mortality [15,16,17]. In part, an obesity-induced accelerated loss of skeletal muscle function with increasing age may act to catalyse poor health outcomes, given a healthy skeletal muscle phenotype is needed to regulate metabolism and for physical activity. Such effects may not solely be associated with altered locomotor muscle function. An obesity-induced accelerated loss in diaphragm function in older age, as reported by Hill et al. [45] in mice, may impair respiratory function, physical activity, lipid oxidation and contribute to the associated higher risk of respiratory disease in old obese individuals [142].

## 9. Future work and Potential Target Therapeutic Strategies for Intervention

As previously outlined, research examining the combined effects of obesity and ageing on skeletal muscle function is sparse and more work is needed to further our understating in this area. Whilst ageing results in the loss of isometric and concentric muscle function, eccentric muscle function is relatively well preserved [47]. The combined effects of ageing and obesity on eccentric function are yet to be explored. Eccentric muscle activity is important for deceleration, stabilisation and absorbing impact [143], thus is imperative for the safe completion of activities of daily living and in mitigating musculoskeletal injury. Obesity effects on eccentric muscle function may, like in ageing, not be analogous to those seen for concentric and isometric function. Mechanistically, eccentric force production differs from other contractility modes and is believed to be reliant on a high rate of reattachment of stretched cross-bridges [144] and on the active stiffening of the giant protein titin [145,146,147]. Obese individuals likely have a greater reliance on high-intensity eccentric contractions given the requirement to control a larger load. These physiological and biomechanical differences provide a rationale for further investigation.

Furthermore, short-term muscle disuse is associated with a catabolic crisis, likely resulting in an accelerated trajectory of functional decline in older adults [148]. For example, short-term bed rest (< 10 days) has been shown to decrease muscle mass and strength in young and older adults [149,150]. Limb suspension and immobilisation in rodent models support this idea [151,152,153] but have further indicated that the recovery of mass and strength in old muscle following reloading is prolonged [154]. Such changes in muscle function likely have profound effects on the safe completion of activities of daily living, and these consequences may be exaggerated in old obese individuals (Figure 2). Obesity may aggravate the age-induced reduction in myogenesis [23,77], resulting in a more substantial catabolic crisis than that seen with ageing alone. Furthermore, impaired myogenesis may account for the prolonged recovery from skeletal muscle damage seen in both HFD and ageing rodent models [155,156]. Future work should consider developing an understanding of the combined effects of obesity and ageing on catabolic crisis models and injury recovery, particularly given the elevated risk of chronic health conditions in the older population.

Given the potential for the deterioration of muscle function to be exaggerated in older obese adults, effective therapeutic strategies to offset these detrimental effects should be considered. In older adults, there is solid evidence for the benefits of resistance training on skeletal muscle function for both old [157,158] and old obese adults [159,160]. Whilst both calorific restriction and exercise are regularly prescribed as a means to reduce obesity, these interventions have poor adherence [161]. In fact, Fildes et al. [162] demonstrated that current nonsurgical obesity treatment strategies are failing to achieve sustained weight loss in the majority of obese patients, with data indicating that the annual probability of patients with obesity attaining a normal body weight being < 1%. Calorific restriction is the most effective method of reducing body mass [163]. However, concurring loss of both adiposity and lean mass may occur [164] potentially resulting in a further reduction in muscle performance [165] a particular risk for older adult populations. Ensrud et al. [166] indicated that intentional weight loss in overweight older women increased the risk of hip fracture, which may be linked to both a reduction in lean mass and a reduced bone mineral density through calorie restriction [167]. Furthermore, recent work has indicated that while calorific restriction may be effective at reversing HFD-induced changes in skeletal muscle metabolism of zebrafish, such an intervention may not be sufficient to reverse changes in contractile performance [72]. Whilst there is evidence to support the effectiveness of well managed dietary interventions in older adults [168], it is not clear if the risks associated with calorie restriction in older adults outweigh the benefits [169]. Conversely, a progressive programme of physical activity may be beneficial for managing body composition [163,164,170] and maintaining muscle function [159,160]. However, poor adherence indicates a demand for low-cost alternatives strategies to improve muscle function. Such outcomes may be achieved via chronic supplementation of dietary supplements. Although not an exhaustive list, dietary supplementation of vitamin D and Resveratrol have shown promise, but are yet to be robustly explored for their potential to reduce or reverse the additional negative impact of obesity on the age-related decline in muscle function. There is more robust support for benefits of dietary protein supplementation to combat muscle ageing, but its effectiveness in the old obese population is not yet established. Below, we consider evidence that may stimulate further study of these dietary supplements in the treatment of muscle weakness in the older obese person.

### 9.1. Vitamin D

Vitamin D (or in its biologically active form 1α,25(OH)2D3, 24R,25-dihydroxyvitamin D3 [24R,25(OH)2D3]) has been demonstrated to play an important role in skeletal muscle function, with vitamin D deficiency associated with a reduction in the force-producing capacity of skeletal muscle [171,172,173], increased fall risk [173,174,175] and a reduction in physical function [173]. Prolonged supplementation of vitamin D has been shown to improve skeletal muscle function in older adults [176,177], and in young healthy and young athletic individuals [178,179,180,181], irrespective of weight status. Such effects are largely apparent in individuals that demonstrate a vitamin D deficiency. Obesity has been shown to cause vitamin D deficiency [182,183], attributed not only to altered behaviour and lower dietary intake but also to reduced synthesis, a reduced intestinal absorption, altered metabolism and elevated accumulation in FM [184,185]. As such, vitamin D deficiency may mechanistically account for at least some of the obesity-associated decrements in muscle function, thus potentially making supplementation an appropriate target for intervention. Through interaction with vitamin D receptors, which are expressed ubiquitously including in skeletal muscle [182], vitamin D likely evokes both systemic and direct muscle effects which may account for the improvement of skeletal muscle function seen in vivo with vitamin D supplementation. There is evidence that vitamin D improves mitochondrial function, muscle insulin signalling, contractile protein synthesis, calcium and phosphate homeostasis and inflammation [184,186,187,188], factors which are imperative for optimal contractile performance. Interestingly, many of these responses counteract the effects elicited by both ageing and HFD in skeletal muscle function [25,27]. Despite some ambiguity, there is also evidence that vitamin D supplementation may reduce body fat in obese individuals [189,190,191]. Low serum 25(OH)D has been associated with elevated parathyroid hormone (PTH) and low intracellular Ca^2+^, which may contribute to obesity due to elevated lipogenesis and suppressed lipolysis [185]. High levels of adiposity, brought about through a positive energy balance, result in elevated intramuscular lipid accumulation [70] and higher visceral fat, both of which are likely major contributors to the HFD-induced reduction in muscle quality. Intuitively, a vitamin D-induced reduction in stored lipids may evoke improved muscle function, attenuating or even to some extent reversing the response previously outlined. Interestingly, vitamin D supplementation has displayed anti-obesogenic effects in mice, where the administration of vitamin D to HFD-induced obese mice halted the progression in body weight gain, hyperglycaemia and hyperinsulinemia induced by diet [192].

### 9.2. Resveratrol

Resveratrol (3,5,4′-trihydroxystilbene), a natural polyphenol compound sourced in trace amounts from grapes, berries and nuts, has received attention for its potential to alleviate some of the health consequences associated with several metabolic diseases, including obesity [193,194]. Supplementation with resveratrol has shown to attenuate some of the detrimental changes that contribute to poor skeletal muscle health in obese individuals [195,196,197]. In obese rodent models, resveratrol has shown to both limit and halt overall, segmental and intramuscular adipose accumulation [195,196,197,198]. These effects have largely been attributed to reduced activity of enzymes involved in fatty acid synthesis and triacylglycerol accumulation in adipose tissue and down-regulated mRNA expression of genes related to the lipogenic pathway [198,199,200]. Furthermore, evidence indicates that resveratrol mitigates obesity-induced low-grade chronic inflammation and reduce AMPK activity [195,197]. Resveratrol may further enhance both size and density of mitochondria [196] and promote myogenesis [201], which may ultimately culminate in improved skeletal muscle contractile function.

Whilst ambiguity exists with respect to the benefits of resveratrol in humans, given the low bioavailability of the compound, it would still appear to elicit positive health benefits [202,203]. The ambiguity in human trials is likely a result of methodological differences between studies, primarily the combination of dose and duration. Ambiguity also exists with the potential therapeutic effects of resveratrol for mitigating declines in muscle mass [204]. Resveratrol provides a protective effect on muscle mass during mechanical unloading in young rodents [205], yet little effect in aged rodents [206,207]. However, recovery of muscle mass following unloading may be enhanced in older muscle when supplemented with resveratrol [206]. Further evidence reports that resveratrol treatment does not reduce an age-related decline in muscle mass [208], but supplementation may be effective in aged muscle when combined with exercise [209]. Whilst further work is needed to assess the efficacy of resveratrol in the context of ageing and obesity, there is evidence to suggest the outlined changes can translate to improved skeletal muscle contractile function and exercise performance across multiple rodent models [196,204,210,211,212,213].

### 9.3. Protein

Given its anabolic and appetite-suppressing effects [214,215,216], dietary protein may be an important macronutrient to combat the additive effects of age and obesity on skeletal muscle function. Optimising the protein or essential amino acids intake in old age has been demonstrated to improve muscle protein accretion [217]. Low protein intake in older adults has been associated with an accelerated loss of lean mass [218,219], grip strength [220] and an enhanced risk of developing mobility limitations [221]. Furthermore, results from meta-analysis indicate that low protein intake is associated with frailty [222]. Such findings justify increasing dietary protein and essential amino acid intake to combat the age-related decline in skeletal muscle function, a topic that has been the subject of several meta-analyses.

When combined with an exercise intervention, primarily resistance training, there is strong support that increasing protein intake in older adults will cause a superior increase in lean mass, leg strength, and some aspects of physical function [223,224,225]. Effects of increased protein intake in the absence of resistance exercise appear controversial. Whilst the results of some meta-analyses indicate increased strength, lean mass and physical function [226,227], others have shown no effects [228,229,230]. The disparity in findings may relate to several factors. It has been suggested that increasing protein intake may have a greater benefit for frail older adults [223,224], those with low habitual protein intake [226] and for men [223]. Furthermore, there is still debate regarding the optimal quantity, distribution and source [231], highlighting a need for more systematic studies.

Irrespective of age, high protein diets have an important role in weight management, attenuating the loss of lean mass that may occur as a result of calorie restricted diets [214,232,233]. Furthermore, the additive effect of protein supplementation and resistance training for increasing lean mass and lower limb muscular strength, reported in a meta-analysis by Liao et al. [223], was equivalent in a high BMI (>30 kg/m^2^) group. Interestingly, however, there is some suggestion that the protein-induced prevention of age-related lean mass loss varies depending on obesity status [234].

Preserved or increased lean mass evokes obvious benefits given its association with muscle function, but there is a need for future work to consider the impact on muscle quality. In particular, focus on branched-chain amino acids such as leucine and β-hydroxy-β-methylbutyrate (HMB), a key metabolite of leucine, have shown promise [235].

### 9.4. Pharmacological Interventions

The aforementioned strategies are relatively cost-effective methods for treating obesity. In some cases, however, traditional methods for treating obesity in young and old adults may not be viable either due to other underlying debilitating factors or poor adherence to a treatment regimen. Pharmacological interventions may therefore be required for improving muscle morphology and contractile function, or the weight status of obese individuals. Many pharmacological treatments are available to treat sarcopenia, targeting improvements in muscle mass, muscle contractile function and performance of activities of daily living [236,237]. However, comparatively fewer pharmacological treatments have been identified for specifically treating older adults with low muscle mass and high FM [238,239]. Given the cross-over between the mechanisms that result in the deleterious effects upon muscle morphology and function, some pharmacological agents are available which have been used to independently treat both sarcopenia and obesity in old age [236,238]. Therefore, the pharmacotherapies considered here will only pertain to those which have been examined in both sarcopenic and old obese populations, namely myostatin inhibition and hormone replacement therapy via testosterone provision.

Myostatin is part of the transforming growth factor beta family which is primarily responsible for negatively regulating muscle mass. A recent review has identified that myostatin may play an important role in contributing to the negative cycle of obesity by not only adversely regulating muscle mass but also reducing lipid oxidation and insulin sensitivity [239]. Myostatin inhibitors and myostatin deletion in rodents has not only shown to increase muscle mass [240,241], but also protect against the negative effects of obesity by preventing insulin resistance and accumulation of body fat, and enhancing fatty acid oxidation [239,242,243,244]. However, studies examining contractile function have reported that myostatin deletion caused the specific tension of both in situ [245] and isolated skeletal muscles to be significantly poorer than normal mice despite inhibition causing significant muscular hypertrophy [240,241]. In humans, very little work has examined the effects of myostatin inhibition in old obese populations, with work primarily focusing on dystrophic adults [246]. One recent study examined myostatin inhibition via the provision of a single dose of bimagrumab to old obese adults which resulted in significant increases in thigh muscle volume and lean body mass, as well as reductions in body mass, whilst muscle strength remained unchanged [247].

Another proposed pharmacological approach is hormone replacement therapy, specifically testosterone replacement [236,238,248]. Testosterone is an androgenic hormone that stimulates muscle anabolism, activates satellite cells [249] and reduces FM [250,251,252]. Ageing is associated with a natural decline in testosterone levels in men, up to 1% per year from the age of 30 [253], which leads to a reciprocal reduction in muscle mass, muscle strength and fat-free mass [254] that is exacerbated in old obese adults with low testosterone [255]. The effects of testosterone replacement in the treatment of sarcopenia and obesity have received extensive coverage in reviews [255,256,257], where many studies report an improvement in muscle mass, reduction in FM and improved skeletal muscle force and power in older healthy men [251,258,259,260,261], obese men [262], and in both young and older women [263,264]. However, some studies in older men report no change in contractile performance following testosterone provision in men with low baseline testosterone [265,266,267]. In the treatment of obesity, many studies focus on targeting men with high FM and low testosterone levels [268]. However, the impact of testosterone administration on muscle mass, muscle contractile function and weight status in old obese adults with low muscle mass and testosterone levels have received little attention. This may be of particular importance given that high visceral adiposity may cause itself reduced testosterone levels [269], thus potentially accelerating the negative cycle of obesity [238].

These pharmacological treatments hold promise for disrupting the negative cycle of obesity by improving body composition via an increase in lean tissue mass and reduction in FM, though the effect on improving contractile function is more equivocal. Future work should aim to identify viable strategies resulting in a safe, but favourable change in all three parameters. Figure 3 provides a summary of the discussed strategies to improve muscle contractile function and weight status in old-obese populations, with focus on the strengths and weaknesses of each strategy.

## 10. Conclusions

The effects of increasing age on skeletal muscle function are well established and there is growing evidence to indicate that obesity (or HFD consumption), independent of ageing, results in detrimental effects on skeletal muscle function. Whilst there is a strong theoretical base that the combined effects of ageing and obesity will accelerate the age-related decline in muscle function, the evidence is controversial, which can largely be accounted for by methodological discrepancies between published works. Despite the ambiguity, some tentative conclusions can be made. Namely, an obesity-induced increase in the maximal force and power-producing capacity of postural muscles may be less apparent in old obese individuals compared to younger counterparts. Ageing in combination with obesity may aggravate the age-related reduction in force to body mass and muscle quality. Such effects may have profound consequences for the safe completion of activities of daily living and the maintenance of a physically active lifestyle. As such, an obesity-induced exacerbated loss in the muscle function of older adults may act as a catalyst for associated negative health outcomes in this population. Further work is needed to better understand the consequences for muscle function, with a particular focus on injury recovery and periods of catabolic crisis. Although dietary interventions to mitigate detrimental effects on muscle function should be approached with caution, there is support for the efficacy of resistance training. Given the poor adherence to exercise programs, this review explored alternative therapeutic strategies for intervention, demonstrating promise for dietary supplementation of vitamin D, resveratrol and protein. Where the need to mitigate these effects might be more urgent, pharmacological strategies may be effective and present an important area for future investigation.

## Figures and Tables

**Figure 1 biomolecules-11-00372-f001:**
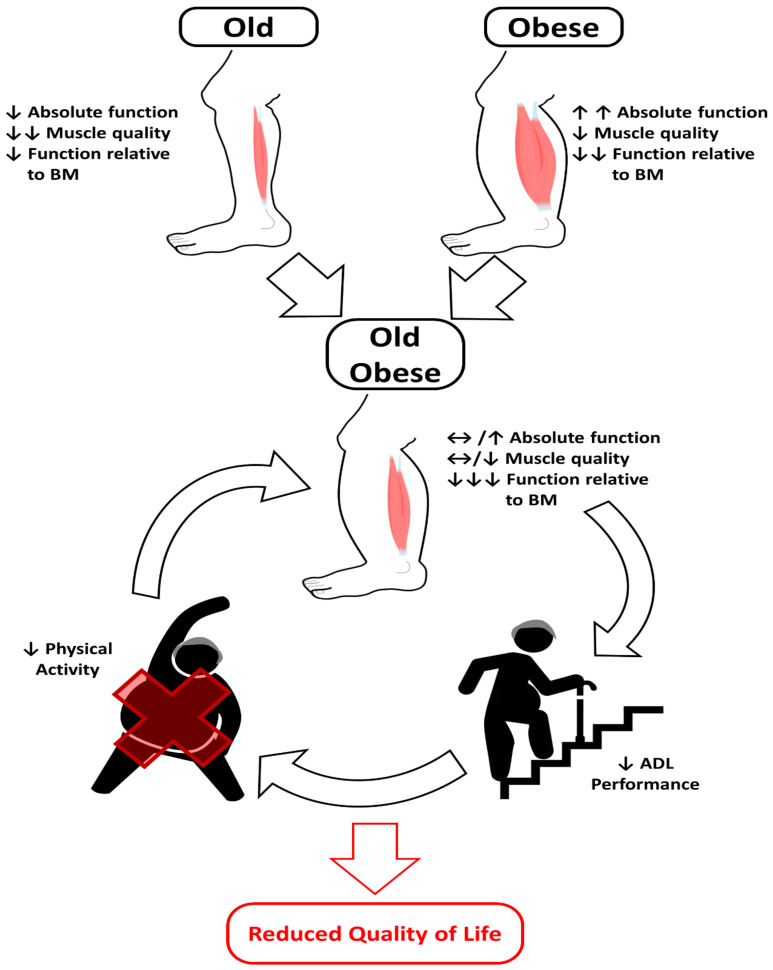
Schema outlining the effects of old age and obesity in isolation on absolute and normalised contractile function and their combined effects on contractile performance in old obese adults. Arrows demonstrate the direction of the effect on contractile performance, where an upwards arrow shows an increase (↑), a downwards arrow a decrease (↓), and a sideways arrow (↔) indicating little change. A greater number of arrows indicates a greater magnitude of the effect. The negative cycle of obesity in old obese adults is demonstrated, resulting in a reduced quality of life.

**Figure 2 biomolecules-11-00372-f002:**
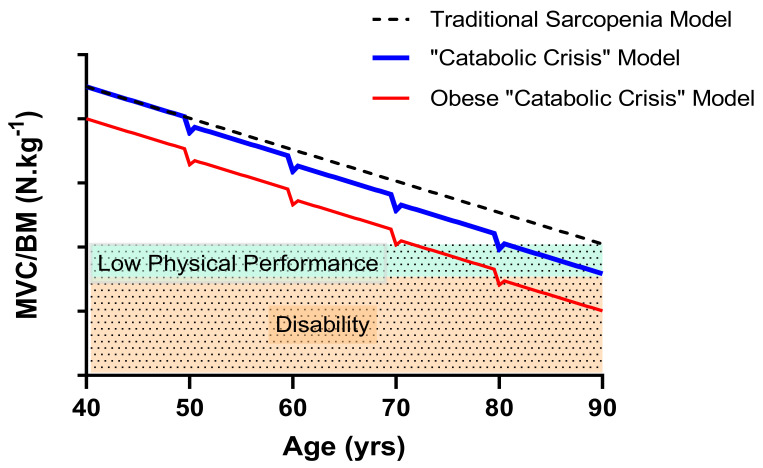
Proposed model of age-related loss in skeletal muscle function relative to body mass accelerated by acute illness or injury. The accelerated loss of function and incomplete recovery has more profound consequences for obese older adults leading to earlier onset of low physical function and disability (MVC, maximal voluntary contraction; BM, body mass; figure adapted from [10,148]).

**Figure 3 biomolecules-11-00372-f003:**
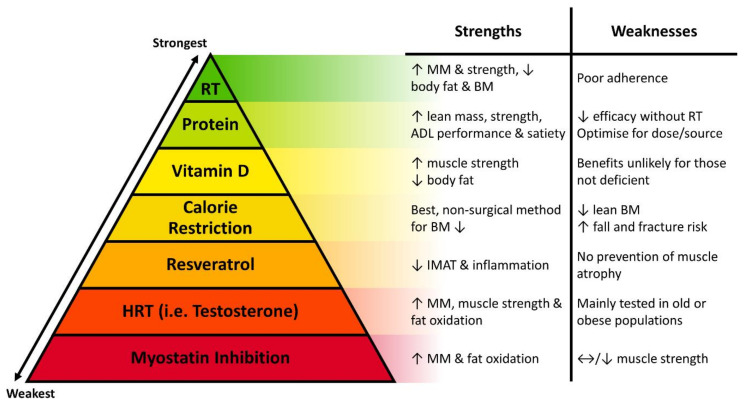
Summary of the strategies used to improve muscle contractile function and body composition in old obese populations. Strategies with the weakest or lowest amount of evidence are in the red segments at the bottom of the pyramid, whilst strategies with the strongest, most abundant evidence are in the green segments at the top of the pyramid. The key strengths and weaknesses attributed to each strategy are listed. Resistance training (RT) and dietary intervention (protein; vitamin D) provide clear benefits for favourable changes in muscle mass (MM), muscle contractile function, body mass (BM), and overall improved performance and completion of activities of daily living (ADL’s). By contrast, pharmacological interventions (resveratrol, hormone replacement therapy [HRT] via testosterone, myostatin inhibition) demonstrate more equivocal benefits in old obese populations. Arrows indicate the direction of change in a particular variable: ↑, increase; ↓, decrease; ↔, no change.

**Table 1 biomolecules-11-00372-t001:** Summary of research examining the effects of obesity on the contractile performance of skeletal muscle in aged populations.

Author	Sex	Participants: Group and Age (Age/Age Range; yrs)	Absolute Contractile Performance	Force to Body Mass Ratio	Muscle Quality	Body Composition and Muscle Morphological Measurements
Miyatake et al. [31]	M and F	Control: Y (20–39) Middle-aged (40–59) O (60–80) Obese: Y (20–39) Middle-aged (40–59) O (60–80)	O IM KE (kg) and HGS (kg) ↔ Y and Middle-aged Obese IM KE ↑ M: Y Obese HGS (kg) ↑ M: Middle-aged Left HGS (kg) ↔ Right HGS (kg) ↑ F: Y Obese Left HGS (kg) ↑ Right HGS (kg) ↔ F: Middle-aged Obese HGS (kg) ↑	Obese IM KE (Kg/kg ^−1^) ↓	N/A	N/A
Pedersen et al. [32]	M and F	All participants (80)	N/A	M: IM TE, TF, EF, HGS and KE (N/ kg ^−1^) ↔ F: IM TE and TF (N/ kg ^−1^) ↔ F: BMI > 29 IM EF, KE and HGS (N/ kg ^−1^) ↓ versus BMI < 24	N/A	N/A
Rolland et al. [99]	F	Lean (80.7 ± 4.1) NW (80.2 ± 3.7) Obese (80.0 ± 3.5)	Obese IM KE (N) ↑ versus lean Obese and NW IM KE (N) ↔ HGS (Nm^2^) ↔ Obese IM EE (N) ↑ versus lean and NW -Sedentary Individuals IM KE (N) ↔ Irrespective of BMI * Active Obese IM KE (N) ↑ versus Active Lean * Active Obese and NW IM KE (N) ↔ *	N/A	N/A	Obese FM and FFM (Kg and %) Total MM, leg MM and arm MM (Kg) ↑
Villareal et al. [35]	M and F	Non-obese non-frail (70.6 ± 0.8) Non-obese frail (77.3 ± 0.5) Obese (76.5 ± 0.9)	Obese IK CON KE and KF (60 s^−1^; N.m) ↓ versus non-obese non-fail Obese and Non-obese frail IK CON KE and KF (60 s^−1^; N.m) ↔	N/A	Obese IK CON KE and KF torque p.u. LE LM (60 s^−1^; N.m/kg ^−1^) ↓	Obese total fat (Kg and %) and FFM (%) ↑ Obese FFM (Kg) ↓
Hilton et al. [36]	M and F	Non-obese (58.0 ± 10.0) Obese (58.0 ± 9.2)	IM DF and PF (N.m) ↓ IK CON PF and DF (60 s^−1^, 120 s^−1^; W) ↓ IK CON PF (60 s^−1^, 120 s^−1^; N.m) ↓ IK CON DF (60 s^−1^; N.m) ↓ IK CON DF (120 s^−1^; N.m) ↔	N/A	IK CON PF p.u. MV (60 s^−1^, 120 s^−1^;) W/cm^3^) ↓ IK CON DF p.u. MV (120 s^−1^; W/cm^3^) ↓ IK CON DF p.u. MV (60 s^−1^; W/cm^3^) ↔	Distal LE IMAT volume (cm^3^) ↑ LE MV (cm^3^), adipose tissue volume (cm^3^) and muscle CSA (cm^2^) ↔
Paolillo et al. [33]	F	Non-obese (54.0 ± 11.0) Obese (58.0 ± 2.0)	IK CON KE (60 s^−1^; N.m) ↔ IK CON KE (300 s^−1^; W) ↑	IK CON KE (60 s^−1^; N.m/kg ^−1^) ↓ IK CON KE (300^−1^; W/kg ^−1^) ↔	IK CON KE p.u. LM (60 s^−1^; N.m./kg ^−1^) ↓ IK CON KE p.u. LM (300 s^−1^; W/kg ^−1^) ↔	BF (%), LM (kg), FM (kg) ↑
Choi et al. [34]	M and F	NW (70.0 ± 2.0) ** Obese (69.0 ± 2.0) **	IK CON KE (N.m) ↑ Type I single-fibre power (µN. FLs^−1^) ↓ Type I fibre maximal shortening velocity (FLs^−1^) ↓ Type I and IIa maximal Ca^2+^ activated force (mN) ↓ Type IIa single-fibre power (µN.FLs^−1^) ↔ Type IIa fibre maximal shortening velocity (FLs^−1^) ↔	IK CON KE (N.m/kg ^−1^) ↓	IK CON KE p.u. thigh MV (N.m/cm^3^) ↓ Type I isolated fibre power p.u. fibre size (W/litre fibre) ↓ Type IIa isolated fibre power p.u. fibre size (W/litre fibre) ↔ Type I and IIa maximal Ca^2+^ activated force p.u. CSA (kN/m^2^) ↓	Total thigh volume (cm^3^), thigh fat volume (cm^3^), thigh MV (cm^3^), intramuscular fat volume (cm^3^), type I and IIa fibre CSA (μm^2^), type I fibre intramyocellular lipid ↑
Tomlinson et al. [100]	F	Y: UW (23.0 ± 6.7) NW (23.2 ± 7.9) OW (23.6 ± 8.0) Obese (30.9 ± 10.7) O: UW (63.8 ± 5.7) NW (63.5 ± 7.7) OW (68.2 ± 4.8) Obese (62.5 ± 9.0)	Y Obese Net IM PF and IM PF (N.m) ↑ versus Y NW and UW Y Obese and OW Net IM PF and IM PF (N.m) ↔ O Net IM PF and IM PF (N.m) ↔ Y IM DF (N.m) ↔ O Obese IM DF (N.m) ↑ Y activation and co-contraction (%) ↔ O activation and co-contraction (%) ↔	Y Obese IM PF (N.m/kg ^−1^) ↓ O Obese IM PF (N.m/kg ^−1^) ↓ versus O UW O Obese, OW and NW IM PF (N.m/kg ^−1^) ↔ Y Obese Net IM PF (N.m/kg ^−1^) ↓ versus Y NW and UW Y Obese and OW Net IM PF (N.m/kg ^−1^) ↔ O Obese Net IM PF (N.m/kg ^−1^) ↓ versus O NW O Obese, OW and UW Net IM PF (N.m/kg ^−1^) ↔	N/A	Obese BF (%), total BF and LM (kg) leg FM (kg) ↑ O Obese Leg LM (kg) ↑ versus O NW, UW and OW Y Obese Leg LM (kg) ↑ versus Y NW and UW Y Obese and OW Leg LM (kg) ↔
Tomlinson et al. [101]	F	Y (25.5 ± 9.0): UW NW OW Obese O (64.8 ± 7.2): UW NW OW Obese	Obese Net IM PF (N.m) ↑ versus NW and UW Obese and OW Net IM PF (N.m) ↔ High BF Net IM PF (N.m) ↑ versus normal BF	N/A	Obese Net IM PF p.u. MV (N.m/cm^3^) ↓ versus NW Obese, UW and OW Net IM PF p.u. MV (N.m/cm^3^) ↔ Obese GM-specific force (GM fascicle force/PCSA) ↓ versus NW and UW Obese and OW GM-specific force (GM fascicle force/PCSA) ↔ High BF GM-specific force (GM fascicle force/PCSA) and Net IM PF p.u. MV (N.m/cm^3^) ↓ versus normal BF	Obese MV (cm^3^) ↑ High BF MV (cm^3^) ↑ versus normal BF
Tibana et al. [102]	F	Non-obese (68.0 ± 6.2) Obese (66.5 ± 5.0)	Leg press, bench press (kg) ↔ Bicep curl (kg) ↑	N/A	N/A	Obese WC (cm), NC (cm), W:H, BF (% and kg), FFM (kg) ↑
Erskine et al. [103]	MandF	Y: Normal BF (24.0 ± 8.4) High BF (28.9 ± 9.7) O: Normal BF (65.5 ± 8.0) High BF (66.0 ± 7.3)	High BF IM PF (N.m) ↑ IK CON PF (60 s^−1^; N.m) ↔ High BF GM fascicle force (N) ↑ High BF activation capacity (%) ↓	High BF IK CON (60 s^−1^) and IM PF (N.m. kg ^−1^) ↓	GM-specific force (GM fascicle force/PCSA) ↔ High BF IK CON (60 s^−1^) and IM PF p.u. MV (N.m/cm^3^) ↓ Y normal BF IK CON (60 s^−1^) and IM PF p.u. MV (N.m/cm^3^) ↑ versus all other groups	GM fascicle length ↔ - High BF GM FPA, FM (kg), LM (kg), GM volume (cm^3^), GM PCSA (cm^2^)↑

Abbreviations: BMI, body mass index; M, male; F, female; Y, young; O, old; NW, normal weight; UW, underweight; OW, overweight; IM, isometric; IK, isokinetic; CON, concentric; KE, knee extensors; EE, elbow extensors; EF, elbow flexors; TF, trunk flexors; TE, trunk extensors; PF, plantar flexor; DF, dorsi flexor; GM, gastrocnemius medialis; HGS, hand grip strength; FM, fat mass; FFM, fat-free mass; MM, muscle mass; MV, muscle volume; BF, body fat; LM, lean mass; LE, lower extremity; IMAT, intramuscular adipose tissue; CSA, cross sectional area; PSCA, physiological cross sectional area; FPA, fascicle pennation angle; W, watts; WC, waist circumference; NC, neck circumference; W:H, waist to hip ratio; N/A, not applicable; p.u., per unit; Net, sum of maximal torque and co-contraction torque; data presented as the mean ± SD; * contractile function adjusted for physical activity; ** data presented as the mean ± S.E.M; ↓/↑ P<0.05, ↔ no change/difference.

**Table 2 biomolecules-11-00372-t002:** Summary of studies examining the effects of obesity on isolated muscle contractile function in aged rodents.

Author	Animal Information	Dietary Protocol	Experimental Protocol	Absolute Contractile Performance	Muscle Quality (Contractile Parameter Per Unit of Tissue Size)	Body Composition and Muscle Morphology Measurements
Abrigo et al. [111]	M C57BL6/10 mice aged 12 weeks Groups: Control HFD	38-week diet Control calorie (%):fat, 10; CHO, 70; protein, 20 HFD calorie (%): fat, 60; CHO, 20; protein, 20	In vivo: forelimb strength via weightlifting links of mass 15.5–54.1 g In vitro: IM tetanus force of whole TA using stim. freq. 10–150 Hz at room temp.	In vivo strength ↓ in HFD	In vivo strength p.u. body mass ↓ in HFD Tetanic stress p.u. muscle CSA (mN/mm^2^) ↓ in HFD at all stim. freq.	Type IIa distribution (%) ↑ in HFD Type IIb distribution (%) ↓ in HFD Percentage of fibres with a larger diameter (µm) ↓ in HFD
Bott et al. [112]	M C57BL/6 mice aged 20 weeks Groups: Baseline Aged-control HFD	13-week diet Control calorie (%):fat, 10.3; CHO, 75.9; protein, 13.7 HFD calorie (%): fat, 45.3; CHO, 40.8; protein, 13.8	IM twitch and tetanus force of whole SOL and EDL at 25 °C	SOL: Twitch and tetanus activation time (mN/ms) ↔ Twitch relaxation time ↔ Tetanus relaxation time ↓ in HFD compared to baseline EDL: Twitch activation and relaxation time ↑ in HFD compared to baseline Tetanus activation time ↔ Tetanus relaxation time ↓ in HFD compared to baseline	Twitch and tetanus stress p.u. muscle CSA (mN/mm^2^) ↔ across all groups	SOL: Type I, IIa, IIx and IIb CSA (µm^2^) ↑ compared to control and baseline EDL: Type IIa and IIb ↓, type IIx ↑ compared to baseline Type IIa ↓, IIx ↔ and IIb ↓ compared to control
Hill et al. [45]	F CD-1 mice aged 70 weeks Groups: Control HFD	9-week self-selected diet Control calorie (%): fat, 7.4; CHO, 75.1, protein, 17.5 HFD calorie (%): fat, 63.7; CHO, 18.4; protein, 17.9	IM tetanus force; WL power and fatigue resistance of whole SOL, EDL and DIA at 37 °C	Activation and relaxation (ms) ↔ for all muscles IM force (mN) ↔ for all muscles WL power ↑ for HFD soleus and EDL	IM stress p.u. muscle CSA (kN.m^2^) ↔ for SOL and EDL, tendency for ↓ in HFD DIA WL power p.u muscle mass ↔ for SOL and EDL, ↓ for HFD DIA WL fatigue resistance ↔ for all muscles	BM (g), circumference (cm), BMI, gonadal FM (g), and FM:BM ↑ in HFD MM (mg) and CSA (m^2^) ↓ in HFD SOL and EDL MM:BM, ↔ for SOL and EDL
Eshima et al. [46]	M C57BL/6 mice aged 2 months Groups: Control HFD	20-month diet Control calorie (%): fat, 5.6; CHO, 53.8; protein, 22.6 HFD calorie (%): fat, 60; CHO, 20; protein, 20	IM tetanus force of whole SOL and EDL using stim. freq. 1–150 Hz	SOL: IM force (mN) ↔ at all stim. freq. Activation and relaxation time (ms) ↔ EDL: IM force ↓ in HFD at 50–150 Hz	SOL: IM stress p.u. muscle CSA (kN.m^2^) ↔ at all stim. freq. EDL: IM stress p.u. muscle CSA ↓ in HFD at 50–150 Hz	BM (g), Abdominal visceral fat (g), EDL IMCL droplet size (µm^2^) ↑ in HFD SOL and EDL MM (mg) ↔

Abbreviations: M, male; F, female; HFD, high-fat diet; CHO, carbohydrates; TA, tibialis anterior; EDL, extensor digitorum longus; SOL, soleus; DIA, diaphragm; stim. freq., stimulation frequency; IM, isometric; WL, work loop; CSA, cross-sectional area; BMI, body mass index; BM, body mass; MM, muscle mass; FM, fat mass; FM:BM, fat mass to body mass ratio; MM:BM; muscle mass to body mass ratio; IMCL, intramyocellular lipid; p.u., per unit. ↓/↑ *p* < 0.05, ↔ no change/difference.

## Data Availability

The data presented in this study was obtain from the cited sources.

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
