# Peer review of "Age-Related Skeletal Muscle Dysfunction Is Aggravated by Obesity: An Investigation of Contractile Function, Implications and Treatment"

_biomolecules, 2021, doi:10.3390/biom11030372_

Round 1

Reviewer 1 Report

This manuscript is a very comprehensive and well written review describing the relationship between ageing, function of skeletal muscle and obesity. The authors summarized the effects of obesity on the contractile performance of skeletal muscle in aged populations, as well as in rodents. The authors also summarized potential therapeutic targets and strategies for muscle function improvement. I recommend acceptance of this manuscript after correction of the editing mark in line 241.  

Author Response

Response: We thank the reviewer for taking the time to read our manuscript. We are pleased that consider our review suitable for publication. We have corrected the editing mark as suggested.

Reviewer 2 Report

  1. In general, a certain redundancy should be avoided in the manuscript. For example, in the abstract “the present study sought…. detrimental effects.” and in the introduction “here we review the… of obesity in old age” looks similar.
  2. The mentioned objective of the review “1) examine if obesity aggravates….” Is not very appropriate here, since this is not a research article.
  3. It will have an additional impact on the manuscript if a brief mechanism of adipogenesis and myogenesis is included.
  4. Heading 6 “Examining the Synergistic Effects of Ageing and Obesity using Isolated Rodent Skeletal Muscle” It is important to revise this title in a concise manner. Under this heading, the content and support table is quite informative, but it is too much for the readers. The text section should be very concise focusing on the own speculations about these studies.
  5. Page 18, Line number 241, correct the sentence, it is a typo error.
  6. In a review article, a pictorial or graphical representation is more appealing than the text part. In order to improve the quality of the article, the authors should include a few more figures.

Author Response

1.In general, a certain redundancy should be avoided in the manuscript. For example, in the abstract “the present study sought…. detrimental effects.” and in the introduction “here we review the… of obesity in old age” looks similar.

Response: We agree, the final section of the introduction was a repeat of the objectives already outlined in the abstract. As such we have removed this information from the introduction. As suggested, we have also endeavoured to remove redundant text in other sections of the manuscript.

2. The mentioned objective of the review “1) examine if obesity aggravates….” Is not very appropriate here, since this is not a research article.

Response: We agree and have amended the wording of this first objective as suggested.

3. It will have an additional impact on the manuscript if a brief mechanism of adipogenesis and myogenesis is included.

Response: Whilst we value this suggestion, given the now extended length of the review, we believe it important to keep this section brief to keep the review focussed on the impact of obesity in old age on muscle function. As adipogenesis and myogenesis have already been considered previously, here we provide a concise account of mechanisms underpinning the obesity-induced changes in muscle function and have now made reference to further literature, including a review by Kalinkovich & Livshits (2017) (Page 4; Line 154) should the reader wish to explore this further.    

4. Heading 6 “Examining the Synergistic Effects of Ageing and Obesity using Isolated Rodent Skeletal Muscle” It is important to revise this title in a concise manner. Under this heading, the content and support table is quite informative, but it is too much for the readers. The text section should be very concise focusing on the own speculations about these studies.

Response: We have revised the title of this section to ‘6. Effects of Ageing and Obesity on Contractile Function of Isolated Rodent Skeletal Muscle’ We have also revised the title of the section prior to ‘5. Effects of Ageing and Obesity on Skeletal Muscle Contractile Function’ for consistency.

We are pleased that you agree with the interest in the work presented in this section. Work examining the function of isolated skeletal muscle has been important in furthering our understanding of muscle contractile performance across several disciplines. There appears, however, to be limited critical appreciation of such work that is needed to better interpret, circumventing the bias of neuromuscular transmision of in situ  preparations, the impact of obesity on skeletal muscle function per se. This critical account of the available literature considering the combined effects of ageing and obesity on isolated skeletal muscle is an important novel point of the review. As such, we believe it is important to keep the narrative but have reduced the length of this section to improve readability.

5. Page 18, Line number 241, correct the sentence, it is a typo error.

Response: We have amended this typographical error

6. In a review article, a pictorial or graphical representation is more appealing than the text part. In order to improve the quality of the article, the authors should include a few more figures

Response: Thanks for this suggestion to enhance the communication of the key points of the review. We have added two figures, one where we consider the impact of obesity in old age on contractile function (figure 1), and another where we visually demonstrate the efficacy of the reviewed strategies for treating obesity in old age (figure 3). We hope these figures, along with the figure visualising catabolic obesity (figure 2) adequately improves the manuscript.

Reviewer 3 Report

The present review article by Tallis et al. summarized the effect of obesity on aging-induced skeletal muscle dysfunctions. The point of view is in line with the time and fine. However, it is just a summary of previous research and lack of originality. The authors should be more clearly about what they want to insist in this review. Other concerns are listed below.

  1. Page3 2nd para. Should define the difference between “muscular power” and “muscular strength”, if it is used in a different meaning.
  2. The authors use the word “synergistic” in this review, but as long as the reviewer read the cited references, there was no study examining the synergistic effect of aging and obesity. Most research investigated the muscle function changes in aged and obese people or the effect of obesity (HFD) on muscle functions in aged people and rodents. So, it’s not proper to use “synergistic”. “Additive”, “aggravate”, “exacerbate” etc. may be suitable.
  3. Table 1 is pretty hard to understand because various results are listed without being organized. The authors should devise the contents and described method of the table.
  4. Abrigo et al [106[ and Bott et al [107] should not be included in Table 2 because the studies did not use aged rodents.
  5. For therapeutic strategies, amino acids treatment is considered to be important in addition to vitamin, resveratrol, myostatin etc. Consideration about amino acids should be included.
  6. Section 8 is long and should be subtitles: Vitamin D, Resveratrol, etc.

Author Response

The present review article by Tallis et al. summarized the effect of obesity on aging-induced skeletal muscle dysfunctions. The point of view is in line with the time and fine. However, it is just a summary of previous research and lack of originality. The authors should be more clearly about what they want to insist in this review. Other concerns are listed below.

Thank you for your comments. We have now included some additional figures to highlight the key-points and illustrate the concepts we have developed in the manuscript. We hope that these additions are what the reviewer hoped to see in the review.

1. Page3 2nd para. Should define the difference between “muscular power” and “muscular strength”, if it is used in a different meaning.

Response: Muscular strength and power have now been defined (Page 2; Lines 65-66).

2. The authors use the word “synergistic” in this review, but as long as the reviewer read the cited references, there was no study examining the synergistic effect of aging and obesity. Most research investigated the muscle function changes in aged and obese people or the effect of obesity (HFD) on muscle functions in aged people and rodents. So, it’s not proper to use “synergistic”. “Additive”, “aggravate”, “exacerbate” etc. may be suitable.

Response: Thank you for the suggestion. We have removed the term ‘Synergistic’ and its derivatives from the manuscript. We have used the terms “additive”, “aggravate”, “exacerbate” and “combined” as proposed.

3. Table 1 is pretty hard to understand because various results are listed without being organized. The authors should devise the contents and described method of the table

Response: We agree that upon reflection Table 1 may have been difficult to follow. As such, we now present a more concise version that we hope more clearly conveys the key findings.

4. Abrigo et al [106[ and Bott et al [107] should not be included in Table 2 because the studies did not use aged rodents.

Response: We share your concerns with respect to the age of the rodents used in these studies. We have text outlining such limitations on Page 5; Line 230-239 of the manuscript. However, we believe the inclusion of these papers is important for the table and the corresponding narrative. Whilst 33- and 50-week old animals are younger than the ages in the majority of rodent studies that examine the effects of ageing on skeletal muscle, there is evidence to suggest that 30 and 50 weeks of age represents an early-ageing response. We have now included this in the manuscript (Page 5; Line 234). Furthermore, the study by Bott et al is titled ‘Musculoskeletal structure and function in response to the combined effect of an obesogenic diet and age in male’ and the study aims are constructed in a way to reflect this. Therefore, we believe it appropriate to include these papers in the table. The text in the manuscript highlights that these papers may represent the impact of obesity even in pre-sarcopenia. Given that the animals used in the study by Abrigo et al are older than that used by Bott et al, for completeness, this study is also included.

5. For therapeutic strategies, amino acids treatment is considered to be important in addition to vitamin, resveratrol, myostatin etc. Consideration about amino acids should be included.

Response: We thank the reviewer for this valuable suggestion. We agree that protein supplementation represents an important therapeutic strategy and have added a section regarding this on Page 11-13; Lines 482-511. 

6. Section 8 is long and should be subtitles: Vitamin D, Resveratrol, etc.

Response: We have added subtitles to this section as suggested.

Round 2

Reviewer 2 Report

The authors have responded satisfactorily to the issues raised